# Neural Atlas Graphs for Dynamic Scene Decomposition and Editing

**Jan Philipp Schneider**[1,2]     **Pratik Singh Bisht**[1]     **Ilya Chugunov**[2]

**Andreas Kolb**[1]          **Michael Moeller**[1,3]     **Felix Heide**[2,4]

[1]University of Siegen     [2]Princeton University     [3]Lamarr Institute     [4]Torc Robotics

## Abstract

Learning editable high-resolution scene representations for dynamic scenes is an open problem with applications across the domains from autonomous driving to creative editing – the most successful approaches today make a trade-off between editability and supporting scene complexity: neural atlases represent dynamic scenes as two deforming image layers, foreground and background, which are editable in 2D, but break down when multiple objects occlude and interact. In contrast, scene graph models make use of annotated data such as masks and bounding boxes from autonomous-driving datasets to capture complex 3D spatial relationships, but their implicit volumetric node representations are challenging to edit view-consistently. We propose Neural Atlas Graphs (NAGs), a hybrid high-resolution scene representation, where every graph node is a view-dependent neural atlas, facilitating both 2D appearance editing and 3D ordering and positioning of scene elements. Fit at test-time, NAGs achieve state-of-the-art quantitative results on the Waymo Open Dataset – by 5 dB PSNR increase compared to existing methods – and make environmental editing possible in high resolution and visual quality – creating counterfactual driving scenarios with new backgrounds and edited vehicle appearance. We find that the method also generalizes beyond driving scenes and compares favorably - by more than 7 dB in PSNR - to recent matting and video editing baselines on the DAVIS video dataset with a diverse set of human and animal-centric scenes. Project Page: https://princeton-computational-imaging.github.io/nag/

## 1   Introduction

There has been growing demand in graphics and vision for high-fidelity static 3D [27, 16] and dynamic 4D [35, 38] reconstruction models, and in particular for *editable* representations which decompose the scene into semantically meaningful components [43]. For autonomous driving, where large collections of labeled video data are required to train driving behavior [13], editable scene representations offer a direct approach to simulate counterfactual driving scenarios – removing, re-timing, or repositioning vehicles and pedestrians to generate new trajectories, or editing the visual elements of the scene to reflect new environmental conditions. This enables systems to expand a limited collected real-world dataset into a richer and more diverse training set while preserving photo-realism and semantic consistency.

In this setting, neural scene graphs [29] have emerged as a versatile hierarchical scene representation [19, 39, 45, 22]; providing structured, object-centric models that enable repositioning and re-rendering elements with high visual quality. These methods model the scene as set of connected nodes – e.g., vehicles, pedestrians, backgrounds – represented as individual radiance fields [9] or collections of Gaussians [4]. The nodes are composited to render views of the scene, optimized during test time to fit an input driving sequence. Beyond visual data, these neural scene graph models can also ingest LiDAR and bounding box information, readily available in autonomous driving datasets [36, 47],

39th Conference on Neural Information Processing Systems (NeurIPS 2025).

to better localize objects in 3D space over time. While nodes in the scene graph can be removed, rotated, or translated while preserving 3D view consistency, directly modifying their appearance is significantly more challenging. This requires altering the underlying Neural Radiance Fields (NeRF) or 3D Gaussian Splatting (3DGS) models, which requires a method to propagate 2D edits into 3D space [18] to preserve view-consistency.

Neural atlas representations [15], a parallel line of work primarily focused on video editing, offer an alternative approach for this kind of appearance manipulation. These methods learn to map a time-varying 3D environment to a set of static lower-dimensional 2D "atlases", analogous to traditional UV unwrapping [12] of an object surface. During the process of fitting the scene, these models can disentangle a foreground object and its visual effects – e.g., the shadows it casts – from its background. These atlases are then *edited like a regular raster image*, with changes to object textures propagating correctly through the video [20]. Unlike neural scene graphs, however, these models do not represent scene elements explicitly in 3D space, and do not have an ordering between them – i.e., they cannot distinguish if one object is in front or behind another. When there are multiple overlapping objects in motion, neural atlas approaches resolve this by learning multiple non-overlapping alpha masks, with "ordering" achieved by the foreground mask cutting a hole out the background mask [24]. While this does not pose a problem for editing videos with a single primary subject, for settings such as driving scenes, this makes it impossible to remove or reposition overlapping vehicles without introducing visual artifacts.

In this work, we introduce *Neural Atlas Graphs* (NAG) as a hybrid high-resolution representation without these limitations. Given input 3D bounding boxes and segmentation masks from autonomous driving stacks, a NAG represents each scene element as a 2D plane with a 3D trajectory through space and time. Each of these planes acts as an independent neural atlas, capturing object motion, parallax, and lighting effects in a view-dependent neural field model. By explicitly modeling object depths and ordering, a NAG allows for flexible appearance editing at high resolution – directly propagating changes from the 2D atlases to the reconstructed video – without introducing distortions between occluded layers. Designed as an inverse problem, the object trajectories and appearance fields are learned jointly, using ray-casting and efficient ray-plane intersections to accumulate colors and opacities along each ray. By fitting to recorded high-resolution images, an accurate scene decomposition evolves naturally based on the varying motion patterns and provided masks. This enables our approach to perform visually consistent removals, additions, rearrangements and texture editing of scene elements in complex, multi-object environments.

We validate the proposed method on automotive scenes [36] and confirm that the method outperforms recent object-specific 3DGS baselines in visual quality by almost 5 dB PSNR on overall scene quality, and up to 11.2 dB PSNR for dynamic objects. This confirms that the method is able to learn accurate scene representations even under fast object motion, diverse reflections or non-rigid motion patterns, while keeping positional and textual editability. For diverse outdoor scenes, with significantly less geometric prior available, covering various (non-)rigid actors, the method performs favorably to recent matting and neural atlas approaches with 7.3 dB PSNR margin, confirming the generality of our approach.

## 2 Related Work

**Video Layer Decomposition**  Representing videos as a composition of individually deforming layers is a long-studied problem [40] with roots in seamless video editing [2, 14]. While more recent works [46, 24] achieve this via trained optical flow networks [37] and UNet-based masking [34], the core principles remain the same: estimate alpha masks and motion for dynamic elements to separate them from a more static background for individual editing and re-composition. Atlas-style methods – e.g., *Unwrap Mosaics* [32] and later *Layered Neural Atlases* [15] – learn 2D-to-2D warps that map scene points onto an unwrapped canvas, similar to a UV map [12]. Recent works explore neural field [15] and neural radiance field [21] scene representations, in which compact networks are optimized at test time to map continuous coordinates in the input video sequence to view-consistent color. However, existing approaches assume videos that consist of a primary object (and its visual effects, e.g., shadows) set against a relatively static background, limiting their effectiveness in complex, multi-object scenes. When multiple objects overlap, these methods rely on single-layer masks, resulting in visual holes where foreground elements cut into background

objects [21]. While *Generative OmniMatte* [20] presents a method to generate realistic content to fill these occlusions, it reconstructs this content in the background canvas and does not model interactions between overlapping dynamic objects. In this work, we propose *Neural Atlas Graphs* with neural atlas representations that *explicitly model multiple interacting layers*, where a point in the scene can be mapped simultaneously to multiple time-consistent objects, enabling robust editing and re-composition even under complex occlusion scenarios.

**3D Dynamic Scene Models**  Implicit representations such as NeRFs [27, 28, 1] and explicit representations such as 3DGS [16, 11, 48] both enable high-fidelity photorealistic reconstructions of static scenes, and can be extended to fit dynamic scenes via learned deformation and flow fields [31, 35, 43]. However, in either case, editing the scene — and propagating those edits in a view-consistent manner – proves highly non-trivial [8, 41, 9] as these representations require changes to be carefully localized to avoid editing the wrong part of the scene, e.g., editing the subject and not their background. Neural Scene Graphs [29, 45] address this by factorizing the scene into per-object radiance fields, treated as graph nodes, enabling simple repositioning or removal of individual objects via edits to their corresponding nodes. This is made possible partially by the structured predictions of modern driving stacks [26] and annotations offered by autonomous driving datasets such as KITTI [10], nuScenes [3], and Waymo [36], which in addition to image data offer LiDAR point clouds, depth maps, 3D bounding boxes, instance, and object or camera trajectories to instantiate graph nodes in 3D space. Recent hybrid methods combine scene graphs with 3DGS [4, 5] to offer more efficient rendering times, but fine-grained editing remains challenging, as changes to individual nodes must be propagated in a view-consistent manner, which remains an open problem. We propose a hybrid representation that builds a 3D scene graph from structured bounding-box, mask, and trajectory data, but models each graph node as a neural atlas [15], allowing both *direct object appearance editing* through manipulation of a 2D canvas, and object repositioning in 3D space.

## 3  Neural Atlas Graphs

Our proposed *Neural Atlas Graph* (NAG) illustrated in Fig. 1 represents the scene as a graph of moving planes oriented in 3D space, with one plane per moving object – e.g., pedestrian, vehicle, bicyclist – plus a background plane. Each plane follows a learned rigid trajectory and carries a surface-aligned optical-flow field together with view-dependent color–alpha maps – capturing non-rigid motion, parallax, and illumination changes. Rendering is done via depth-ordered ray-casting and alpha compositing across the planes. The subsequent sections detail these components.

### 3.1  Representation

Our *Neural Atlas Graph* (NAG) representation takes as input an arbitrary video scene $\mathcal{I} \in \mathbb{R}^{F \times W \times H \times 3}$, comprised of $F$ 3-channel RGB images of size $W \times H$, and a stack of coarse masks $\mathcal{M} \in \{0, 1\}^{F \times W \times H \times N}$ corresponding to $N$-many foreground objects – nodes in our graph. The remaining unmasked region is represented with an additional background node. To establish the initial position, rotation, and ordering of nodes within the scene graph we primarily rely on the orientation of supplied 3D bounding boxes. If 3D bounding boxes are not available, we fall back to homographies determined from monocular depth estimation for plane initialization [44]. Unfortunately, dynamic scene models such as Neural Atlas Graphs (NAG) inherit the same scene and camera motion ambiguities as found in visual Simultaneous Localization and Mapping (vSLAM) problems. Therefore, we also require an initial estimation of the camera extrinsics and intrinsics, similar to common NeRF [27] and 3DGS [16] approaches. We represent each node $\{\mathcal{N}_i\}_{i=0}^N$ as a tuple $\mathcal{N}_i = (c_i, \alpha_i, f_i, g_i, s_i)$ containing color $c_i : [0, 1]^2 \to \mathbb{R}^3$, opacity $\alpha_i : [0, 1]^2 \to \mathbb{R}$, optical flow field $f_i : [0, 1]^2 \times [0, 1] \to \mathbb{R}^2$, time-dependent position $g_i : [0, 1] \to \mathbb{R}^{4 \times 4}$ and plane size $s_i \in \mathbb{R}^2$.

### 3.2  Image Formation

NAG are rendered with a forward ray-projection model. Each pixel intensity at a timestep $t \in [0, 1]$ in the image $\mathcal{I}$ is composed by aggregating radiance at typically a handful of plane/ray intersections for rays with direction $d \in \mathbb{R}^3$ and origin $o \in \mathbb{R}^3$ (see supplementary material for details). Given a planar node at position $g_i$, decomposable into a position $p$ and normal vector $n$, and the ray $r(l) = o + ld$,

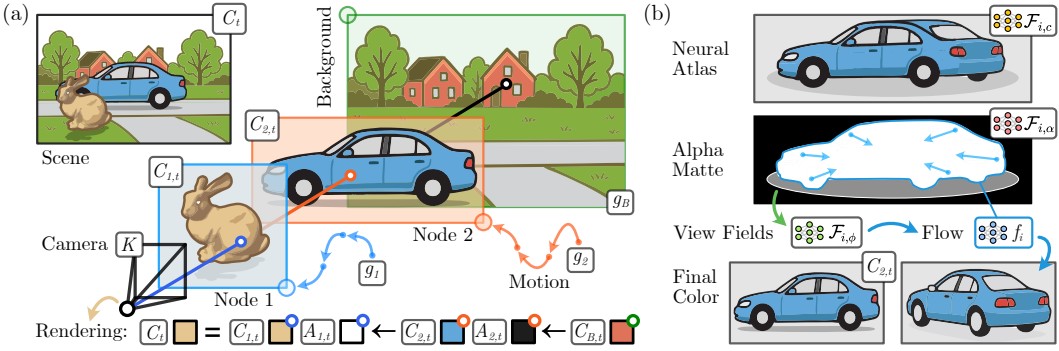

Figure 1: Neural Atlas Graphs - A NAG represents dynamic scenes (a) as a graph of moving 3D planes (one per object/background). Each plane undergoes rigid transformations and encodes view-dependent appearance/transparency using neural fields $\mathcal{F}$ (b) along a learned trajectory $g_i$. The planar optical flow $f_i$ models non-rigid motion and parallax, while learning the representation and rendering is done via opacity-weighted ray casting of $C_{i,t}$, $A_{i,t}$ using position based z-buffering.

the ray-plane intersection $l = (p - o) \cdot n/(d \cdot n)$ may yield an intersection point[1] $x_{\text{world}} = o + ld$. When projected to a finite sized plane $i$, resulting in $x \in [0,1]^2$ (Sec. 3.3), and applied for each node in the NAG, the object color $C_{i,t}$ and opacity $A_{i,t}$ at time $t$ can be determined via a planar optical flow field $f_i$:

$$C_{i,t} = c_i(x + f_i(x,t)), \ A_{i,t} = \alpha_i(x + f_i(x,t)). \tag{1}$$

We alpha-composite [17] each plane intersection along the ray, yielding the final per-pixel color $C$

$$C = \sum_{i=0}^{N} C_{i,t} A_{i,t} \prod_{j=0}^{i-1} (1 - A_{j,t}), \tag{2}$$

of each ray. Given all object colors $C_{i,t}$ and opacities $A_{i,t}$ have been ordered by the distance to the camera in ascending order. If a ray does not hit the corresponding plane – falling off the edge of the finite extent $s_i$ of the plane – we set $A_{i,t} = 0$.

### 3.3 Parametrization of Neural Atlas Nodes

Next, we describe the parametrization of the model components inside nodes $\mathcal{N}_i = (c_i, \alpha_i, f_i, g_i, s_i)$. We opt for modeling our NAG nodes in a 3D space, where each planar node is assigned a 3D position and orientation with 6 DoF assuming a rigid motion model.

**Rigid Plane Pose [$g_i$]**   The affine rigid transformation matrix $g_i(t)$ encodes the node's trajectory over time. We decompose this matrix into translation $T_i(t)$ and rotation $R_i(t)$ components, which are learned independently. To ensure temporal coherence, we use smooth Hermite splines [7] as

$$T_i(t) = \tilde{T}_{i,t} + \eta_{\text{T}} \cdot S(t, \mathcal{P}_i^{\text{T}}). \tag{3}$$

Here, we learn an offset relative to an initial position $\tilde{T}_i \in \mathbb{R}^{F \times 3}$. When 3D bounding boxes are available, $\tilde{T}_{i,t}$ corresponds to the box center at time $t$.

The function $S : [0,1] \times \mathbb{R}^P \to \mathbb{R}^F$ denotes the piecewise cubic Hermite spline interpolation of $P$ number of zero-initialized, learnable control points $\mathcal{P}_i^{\text{T}} \in \mathbb{R}^{P \times 3}$. The weight $\eta_{\text{T}} = 0.5$ controls the contribution of the spline. Adjusting the number of control points $P$ determines the smoothness of the trajectory. For rotation $R_i(t)$, we use the orientation of the bounding box with an added offset[2]. When bounding boxes are not available, we instead learn the rotation using the following offset model

$$R_i(t) = \tilde{R}_{i,t} \cdot q(\eta_{\text{R}} \cdot S(t, \mathcal{P}_i^{\text{R}})), \tag{4}$$

---

[1]Given non-parallel rays, e.g. $d \cdot n \neq 0$, and assuming $o$ is not within the plane.

[2]The offset is chosen to align the plane with the box's front, side, or diagonal face—whichever yields the smallest inclination to the camera view. This allows planar representations even for turning objects, like cars, as long as their rotation remains under 180°.

given an initial rotation $\tilde{R}_i \in \mathbb{H}^{F3}$, we apply a rotation offset via the rotation-vector-to-quaternion mapping $q : [0, 2\pi)^3 \rightarrow \mathbb{H}$, using zero-initialized learnable offsets $\mathcal{P}_i^R \in \mathbb{R}^{P \times 3}$. In the absence of bounding boxes, both $\tilde{T}_i$ and $\tilde{R}_i$ are estimated by decomposing per-object image homographies [25] into relative 3D translations and rotations. These are then cumulatively applied to a planar projection of a monocular depth estimate, yielding per-frame pose estimates.

**Plane Extent [$s_i$]**  To prevent merging of distinct objects exhibiting similar rigid motion and to improve efficiency, we constrain each plane to a finite extent. A point $x_{\text{world}} \in \mathbb{R}^4$ in homogeneous world coordinates lies on the finite plane of object $i$ if the coordinates of its planar correspondence point $x = (g_i(t)^{-1} x_{\text{world}} + 0.5) \odot [s_x, s_y, 0, 0]^\intercal$ are within [0, 1] range, where $\odot$ denotes element-wise multiplication.

The plane scale $s_i = \{s_x, s_y\}$ is pre-estimated from the largest mask within the stack $\mathcal{M}$, plus a relative margin to incorporate object-associated effects, e.g., shadows, and mask inaccuracies.

**Node Color [$C_{i,t}$] and Opacity [$\mathrm{A}_{i,t}$]**  Given a ray–plane intersection point $x \in [0, 1]^2$ and its spherical view-angle $\phi \in [0, 1]^2$ in the plane coordinate system, we model the color and opacity of each node as the combination of a base color field $\tilde{C}_i : [0, 1]^2 \rightarrow \mathbb{R}^3$ and opacity field $\tilde{A}_i : [0, 1]^2 \rightarrow \mathbb{R}$, each augmented by two separate learned neural fields. The base appearance, estimated via forward projection onto masked regions, is parameterized on a pixel grid and extended to arbitrary $x \in [0, 1]^2$ via bilinear interpolation. We use two types of neural fields: view-agnostic fields $\mathcal{F}_{i,c} : [0, 1]^2 \rightarrow \mathbb{R}^3$ and $\mathcal{F}_{i,\alpha} : [0, 1]^2 \rightarrow \mathbb{R}$, and view-dependent fields $\mathcal{F}_{i,\phi,c} : [0, 1]^4 \rightarrow \mathbb{R}^3$ and $\mathcal{F}_{i,\phi,\alpha} : [0, 1]^4 \rightarrow \mathbb{R}$, the latter implicitly regularized via a coarse-to-fine scheme [4]. This combination balances editability – requiring time-consistent atlas content – and adaptivity to scene dynamics, lighting, and view direction, resulting in both temporal stability and high visual fidelity. Our representation is then

$$c_i(x) = \tilde{C}_i(x) + \eta_c \cdot \mathcal{F}_{i,c}(x) + \eta_\phi \cdot \mathcal{F}_{i,\phi,c}(x, \phi), \tag{5}$$

$$\alpha_i(x) = -\log\left(\frac{1}{\tilde{A}_i(x)} - 1\right) + \eta_\alpha \cdot \mathcal{F}_{i,\alpha}(x) + \eta_\phi \cdot \mathcal{F}_{i,\phi,\alpha}(x, \phi), \tag{6}$$

which we use in (1) to represent $C_{i,t}$ and $\mathrm{A}_{i,t}$. While using the base estimates $\tilde{C}_i, \tilde{A}_i$ allows to precondition the object to learn, the subsequent MLPs $\mathcal{F}_{i,c}, \mathcal{F}_{i,\alpha}$, including positional encodings [28], refine projection errors and learn a temporally consistent, editable representation. To capture view- or time-dependent appearance changes, we introduce an additional MLP $\mathcal{F}_{i,\phi}$, which predicts color and opacity offsets based on the planar point $x$ and its associated spherical view angle $\phi$ at the ray–plane intersection. We weight the contributions of the networks using fixed scalars $\eta_c = \eta_\alpha = \eta_\phi = 0.1$, and enforce valid ranges by clamping color values and applying a sigmoid to opacity, ensuring $c_i(x), \alpha_i(x) \in [0, 1]$.

**Flow Field [$f_i$]**  We rely on a temporally changing flow field $f_i : [0, 1]^2 \times [0, 1] \rightarrow \mathbb{R}^2$ attached to each node to model small non-rigid deformations and depth-induced parallax, c.f. (5). To ensure smoothness, we use a spline-based flow model [6], which shifts the ray–plane intersection point $x$ before it is passed to the subsequent networks. That is

$$f_i(x, t) = \eta_f \cdot S(t, \mathcal{F}_{i,f}(x)). \tag{7}$$

The flow field $\mathcal{F}_{i,f} : [0, 1]^2 \rightarrow \mathbb{R}^{P_f \times 2}$ is implemented as an MLP similar to the color and opacity networks. However, instead of predicting a single flow vector, it outputs a set of $P_f$ control points, which are interpolated using a cubic Hermite spline $S(t, \cdot)$ (cf. Sec. 3.3) to produce a smooth, time-varying flow. We employ a coarse-to-fine fitting strategy by progressively masking the encoding dimensions (see supplementary material). Similar to color and alpha, the flow is modeled as an offset and scaled by a fixed weight $\eta_f = 0.1$.

### 3.4  Background Atlas

In addition to the dynamic foreground objects, we model the background using a dedicated node covering all non-masked regions. The background is represented as a fixed planar atlas with no

---

[3]$\mathbb{H}$ denotes the set of unit quaternions.

[4]We detail the coarse-to-fine scheme, along with our *phase-based learning* within the supplementary material.

rigid transformation or opacity variation. Its global pose $g_B \in \mathbb{R}^{4 \times 4}$ is held constant throughout the sequence and placed behind all object bounding boxes. Its orientation and size are chosen to ensure full visibility across the entire camera trajectory, including rotations and translations. The background has the same color and flow model as the foreground nodes (see (5), (7)), but uses a fixed opacity $A_B = 1$ and omits rigid motion to maintain a consistent reference frame.

### 3.5  Optimization

We fit the model by sampling mini-batches of random pixels and timestamps from the input video as ground truth values $y \in \mathcal{I}$ and render color predictions $\hat{y}$. We jointly optimize the transformation parameters and network parameters of all nodes with the loss

$$\mathcal{L}_{\text{atlas}} = ||\hat{y} - y||_1 + \beta \cdot ||\hat{a} - m||_1, \tag{8}$$

which combines a photometric $\ell_1$ term with a mask loss that compares the predicted opacity $\hat{a}$ to the input mask $m \in \mathcal{M}$. The mask term encourages objects to remain opaque in masked regions while allowing flexibility in unmasked areas to represent shadows or object-induced effects. We empirically set $\beta = 0.005$ to balance these objectives.

### 3.6  Atlas Texture Editing

To edit the appearance of an object $i$ (or the background), we require a user-provided RGBA texture – with color $\hat{c} \in [0,1]^{W \times H \times 3}$ and alpha $\hat{\alpha} \in [0,1]^{W \times H}$ – within the image space of a dedicated reference image $\mathcal{I}_{\text{ref}}$. This texture gets ray-projected onto the planar object $\mathcal{N}_i$. On the plane, we can numerically invert the learned flow model and bilinearly interpolating values, to store the user texture $\hat{c} : [0,1]^2 \to \mathbb{R}^3$ and $\hat{\alpha} : [0,1]^2 \to \mathbb{R}$ on a regular grid in atlas space. This allows us to sample the texture at arbitrary ray-plane intersection points $x$. Further, we can blend the values at $\hat{c}(x)$, $\hat{\alpha}(x)$ with the original color model (5) via alpha matting,

$$c_i^*(x) = (1 - \hat{\alpha}(x)) \cdot c_i(x) + \hat{\alpha}(x) \cdot \hat{c}(x), \tag{9}$$

to our learned object representation. Using $c_i^*$ instead of $c_i$ within (1) allows us to render the scene including the user-provided texture.

Storing the texture representation within atlas space is critical: it ensures the edit is independent of the camera motion (given its definition on the node, which undergoes rigid motion) and enables view-consistent editing of subsequent frames as long as these are accurately modeled within our flow representation. Furthermore, by defining textures in image space and projecting them, we allow for better edit comparability since textures are defined independently of the learned flow, in contrast to [15]. This design also enables the seamless use of standard image editing tools and off-the-shelf image generation models.

## 4  Experiments

To evaluate the efficacy and versatility of the proposed method, we conduct three distinct experiments. First, we assess the method with quantitative and qualitative comparisons to state-of-the-art driving scene reconstruction techniques [4, 43]. Next, we validate and compare the editing capabilities, such as object removal, and texture editing, on challenging autonomous driving scenes. Finally, we assess the generalization of the approach on diverse non-driving outdoor scenes.

### 4.1  Driving Scene Reconstruction

**Setup**   For evaluating the quality of our method in autonomous driving scenarios, we rely on a subset of the Waymo [36] open dataset. We select scenes with small ego motion but many objects, occlusions, and diverse and large object motions. In total, we evaluate on 7 scene segments including up to 199 images each, sampled at 10Hz using the forward camera feed. We divide each data segment into sequences of 21 to 89 images, leading to 25 subsequences to be reconstructed.

We filtered the 3D bounding boxes provided for actually moving objects and used the sparsely provided instance segmentation masks in a semi-automatic process using SAM2 [33] to get reasonable masks for every image. Notably, an exemplary ablation study in the supplementary material indicates that our approach maintains applicability even when using coarser masks or only bounding boxes.

Representative ground truth images of these dataset sequences, including our masks and further training details can be found in our supplementary material or code page: `https://github.com/jp-schneider/nag`.

**Assessment**    We evaluate our method against the recent OmniRe [4] method, an object-centric 3DGS approach with explicit SMPL [23] human modeling and EmerNeRF [43], a recent NeRF method including static and dynamic radiance fields [5]. Tab. 1 reports PSNR, SSIM [42] and LPIPS [49] on the individual scenes. The method consistently outperforms the baselines, reaching a significant 5 dB PSNR mean increase over OmniRe and 7 dB over EmerNeRF. To assess these differences visually, we show comparisons in Fig. 2 but recommend viewing the full-resolution images and videos provided in the supplementary. Especially under rapid motion of objects, OmniRe [4] tends to produce artifacts. This can become apparent in missing or smoothed out objects edges, foots of cyclists, partially visible side-mirrors, missing distant objects or reflections which are more reliably modelled with our approach. EmerNeRF [43] tends to render objects over-smoothed, indicating under-parameterizations due to its scene representation in a static and dynamic radiance field. Our NAG node formulation is able to precisely model fine high-resolution details, such as spinning wheels, pedestrian motion, reflections or objects in the distance, even if they are part of the background plane.

Table 1: Quantitative Evaluation on Dynamic Driving Sequences of the Waymo [36] Open Driving Dataset. Best results are in bold. ORe refers to OmniRe [4], and ERF to EmerNeRF [43].

| Seq. | PSNR ↑ | | | SSIM ↑ | | | LPIPS ↓ | | |
|---|---|---|---|---|---|---|---|---|---|
| | Ours | ORe | ERF | Ours | ORe | ERF | Ours | ORe | ERF |
| s-975 | **40.21** | 37.35 | 34.83 | **0.976** | 0.968 | 0.937 | **0.058** | 0.080 | 0.143 |
| s-203 | **43.15** | 36.93 | 36.07 | **0.978** | 0.966 | 0.936 | **0.070** | 0.094 | 0.205 |
| s-125 | **43.32** | 38.74 | 35.20 | **0.980** | 0.970 | 0.933 | **0.057** | 0.079 | 0.182 |
| s-141 | **42.55** | 36.14 | 34.83 | **0.978** | 0.964 | 0.924 | **0.057** | 0.087 | 0.178 |
| s-952 | **41.89** | 39.67 | 35.32 | 0.976 | **0.977** | 0.938 | 0.058 | **0.050** | 0.120 |
| s-324 | **40.85** | 32.58 | 33.63 | **0.977** | 0.953 | 0.926 | **0.038** | 0.071 | 0.124 |
| s-344 | **41.84** | 36.67 | 35.24 | **0.983** | 0.973 | 0.946 | **0.031** | 0.043 | 0.084 |
| Mean | **41.85** | 36.78 | 34.93 | **0.978** | 0.967 | 0.934 | **0.051** | 0.070 | 0.142 |

We further evaluate the quality of the dynamic objects in isolation, grouped into a rigid "Vehicle" class and a non-rigid "Human" class, which allows us to differentiate between objects that may benefit differently from our underlying rigid-motion model. Tab. 2 validates significant improvements over the best baseline, with PSNR gains of 11.2 dB and 10.7 dB, and SSIM increases of 0.064 and 0.080 for the Vehicle and Human classes, respectively. These findings confirm that the quality of our method does not stem from better background rendering but indeed from the high accuracy of our model in representing even non-rigidly moving actors.

---

[5]For an detailed introduction on our baselines we refer to our supplementary material.

Table 2: Quantitative Evaluation of Human and Vehicle Rendering on Waymo [36] Driving Sequences.

| Seq. | Vehicle PSNR ↑ | | | Vehicle SSIM ↑ | | | Human PSNR ↑ | | | Human SSIM ↑ | | |
|---|---|---|---|---|---|---|---|---|---|---|---|---|
| | Ours | ORe | ERF | Ours | ORe | ERF | Ours | ORe | ERF | Ours | ORe | ERF |
| s-975 | **46.79** | 33.09 | 30.21 | **0.991** | 0.939 | 0.82 | **45.37** | 32.99 | 28.53 | **0.989** | 0.927 | 0.777 |
| s-203 | **41.90** | 30.45 | 27.10 | **0.986** | 0.910 | 0.774 | **45.40** | 34.85 | 33.54 | **0.986** | 0.950 | 0.901 |
| s-125 | **41.00** | 28.72 | 24.55 | **0.989** | 0.878 | 0.709 | N/A | N/A | N/A | N/A | N/A | N/A |
| s-141 | **43.21** | 33.22 | 27.36 | **0.981** | 0.929 | 0.744 | **44.22** | 33.31 | 28.86 | **0.986** | 0.907 | 0.769 |
| s-952 | **40.94** | 31.15 | 27.70 | **0.986** | 0.928 | 0.810 | **40.45** | 32.32 | 28.10 | **0.968** | 0.894 | 0.740 |
| s-324 | **41.71** | 31.03 | 27.87 | **0.986** | 0.921 | 0.798 | **44.12** | 32.09 | 26.40 | **0.988** | 0.894 | 0.689 |
| s-344 | **43.97** | 33.02 | 30.65 | **0.985** | 0.931 | 0.835 | **40.99** | 30.20 | 25.94 | **0.975** | 0.882 | 0.721 |
| Mean | **42.88** | 31.69 | 28.09 | **0.986** | 0.922 | 0.787 | **42.94** | 32.24 | 27.78 | **0.981** | 0.901 | 0.744 |

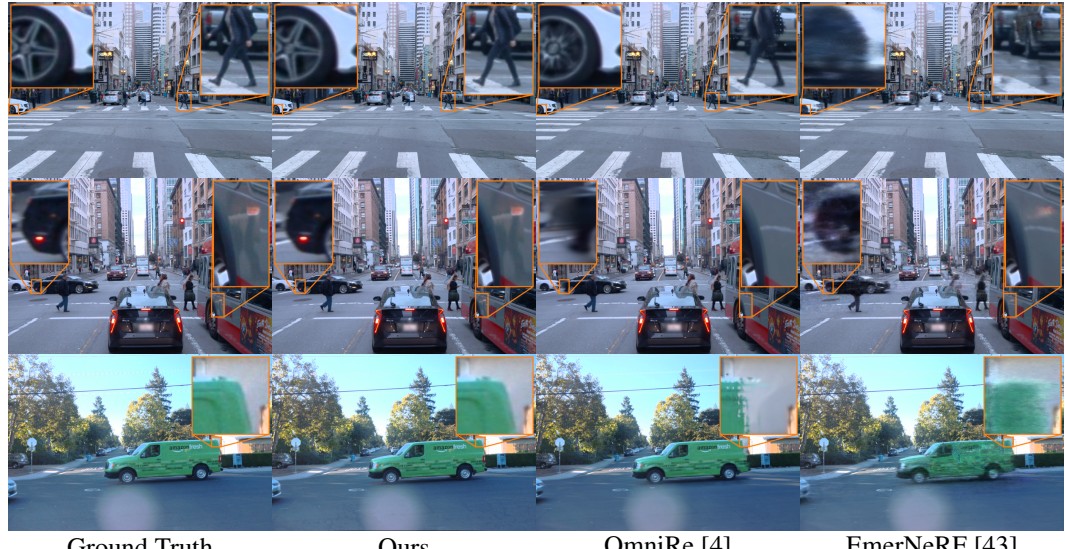

| Ground Truth | Ours | OmniRe [4] | EmerNeRF [43] |

Figure 2: Visual quality comparisons (sequences s-952, 344, 975) show that our method achieves higher fidelity by producing way fewer artifacts on rapid motion (e.g., spinning wheels, edges), significantly reducing motion blur, and preserving finer details like reflections.

**Scene Editing**   We showcase a scene decomposition task in Fig. 3, where all dynamic actors are extracted and separated from the background[6]. Comparing to OmniRe, our method suffers from significantly fewer artifacts around the object peripheries, adding correct shadows and keeping distant information. Given our graph approach, we can further remove, shift, and add additional nodes to the graph and render these into our scene. These functions, along with an evaluation on temporal consistency, are demonstrated in our supplementary material. Also, given our planar nodes, we can take arbitrary textures in image space and project them onto one or multiple nodes to change their appearance along time, as demonstrated in Fig. 4. By combining object removal, and background texture editing, we can generate realistic counterfactual driving scenes. To demonstrate this capability, we execute a realistic editing task within the Waymo s-203 sequence, as shown in Fig. 5. Here, the car integrates naturally with the edited street, correctly incorporating painted speed signs and a zebra crossing. Crucially, the shadow casts realistically along these edited areas, showcasing our method's ability to maintain natural occlusion behavior. Furthermore, the seamless removal of both the person in the foreground and the car in the background, without recognizable artifacts, highlights the precision required for reliable testing within autonomous driving stacks.

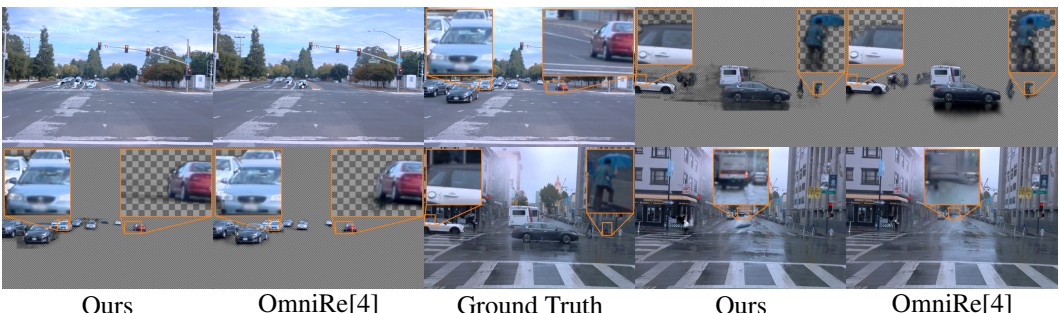

| Ours | OmniRe[4] | Ground Truth | Ours | OmniRe[4] |

Figure 3: Scene Decomposition on Waymo (sequences s-125 and 141). Visual comparison demonstrates our method's capability for accurate object decomposition, preserving sharp boundaries and shadows for both rigid and non-rigid entities. Furthermore, our approach effectively maintains object information even under occlusion and adverse weather conditions.

---

[6]For a clearer illustration, please refer to the supplementary videos.

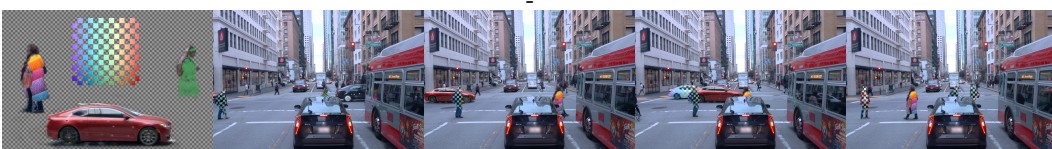

Figure 4: Consistent Appearance Editing. We showcase the consistent application of a textures (left) onto dynamically moving objects, changing their appearance and model occlusions accurately.

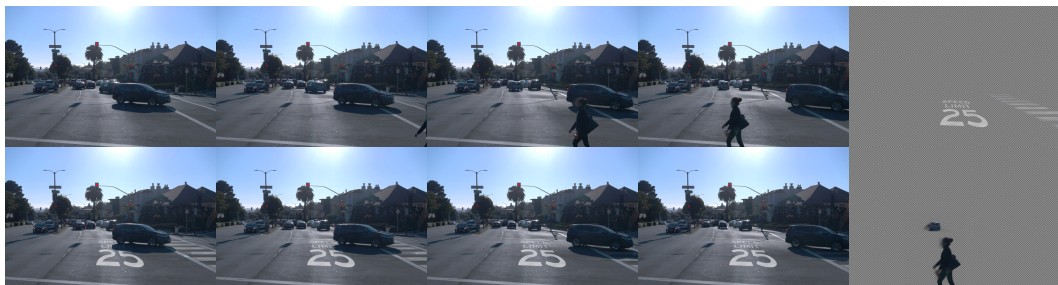

Figure 5: Effective scene manipulation in s-203. Four timestamps from the s-203 scene are shown, with ground truth (top) and our edits (bottom). The applied background texture is presented top-right, and the removed objects are shown bottom-right. The natural blending of the car and its shadow with the edits demonstrates our realistic occlusion handling, showcasing our methods applicability in creating counterfactual driving scenarios.

## 4.2 Outdoor Video Scenes

We validate the generalization of our method on diverse outdoor scenes from the DAVIS dataset [30], which has been regularly used by matting methods [15, 24, 21] and provides high-resolution images (up to $1920 \times 1080$). We evaluate on 15 specific sequences following [15, 21]. To ensure comparability between our method and the baseline approaches, we match the evaluation setups of all methods, see Supplemental Material for details.

We report our results in Tab. 3 and find an average 7 dB PSNR improvement w.r.t the best baseline. While the overall performance varies more than in the automotive scenes, due complex camera and object motion, the method also outperforms all baseline approaches for all sequences. Notably, when objects exhibit rigid-motion behavior (e.g., blackswan, car-shadow or kite-surf), our method achieves its best results. Furthermore, even in scenes involving non-rigid motion (e.g., bear, hike, and elephant), where the challenges are more pronounced, we still outperform the state-of-the-art baselines, validating the generalization to outdoor scenes with various actors. We provide additional baseline information, visual examples, and scene decompositions, in our supplementary material.

## 4.3 Limitations

Our method relies on the assumption that objects and background can be projected onto a plane, imposing limits on camera and object rotation (less than 180 degrees). Correspondingly, Novel View Synthesis and accurate 3D representation is not within the scope of our current manuscript. Accurate initial plane initialization via bounding boxes or geometric priors is recommended, and suboptimal initialization can lead to early failure. Projection errors on non-planar object geometry may accumulate and hinder precise initial position determination.

Representing large camera motion that significantly changes the background is also challenging for the proposed plane assumption, and is further studied in the supplementary material. This is due to the difficulty of capturing large flow vectors, particularly when sampling rays from only a subset of time steps and coordinates as we do. However, we note that the use of view dependency compensates for these errors, although it does come at the cost of reduced texture editability.

Table 3: Quantitative evaluation results on the Davis Dataset [30] of diverse outdoor scenes. Our method consistently outperforms the Layered Neural Atlases (LNA) [15] and OmnimatteRF (ORF) [21] baselines, achieving the best metric scores along all sequences. Also scenes with intricate non-rigid motion where alignment is learned are more demanding, our method still yields superior results, with particularly significant gains in sequences well-suited to our motion model, such as motorbike, kite-surf, and car-shadow.

| Sequence | PSNR ↑ | | | SSIM ↑ | | | LPIPS ↓ | | |
|---|---|---|---|---|---|---|---|---|---|
| | Ours | ORF | LNA | Ours | ORF | LNA | Ours | ORF | LNA |
| bear | **33.47** | 24.88 | 26.51 | **0.934** | 0.658 | 0.771 | **0.091** | 0.464 | 0.287 |
| blackswan | **36.36** | 26.67 | 29.26 | **0.938** | 0.739 | 0.815 | **0.097** | 0.458 | 0.318 |
| boat | **35.83** | 28.63 | 30.15 | **0.932** | 0.761 | 0.816 | **0.099** | 0.376 | 0.274 |
| car-shadow | **36.67** | 29.26 | 28.47 | **0.947** | 0.861 | 0.850 | **0.084** | 0.313 | 0.269 |
| elephant | **33.91** | 26.94 | 28.34 | **0.922** | 0.731 | 0.772 | **0.088** | 0.423 | 0.325 |
| flamingo | **34.96** | 25.74 | 27.10 | **0.928** | 0.753 | 0.783 | **0.106** | 0.483 | 0.349 |
| hike | **29.74** | 25.15 | 24.77 | **0.886** | 0.698 | 0.682 | **0.108** | 0.388 | 0.343 |
| horsejump-high | **34.78** | 28.35 | 27.28 | **0.932** | 0.846 | 0.830 | **0.074** | 0.249 | 0.226 |
| kite-surf | **37.96** | 28.04 | 27.88 | **0.949** | 0.780 | 0.780 | **0.068** | 0.420 | 0.400 |
| kite-walk | **37.96** | 29.44 | 29.58 | **0.941** | 0.804 | 0.818 | **0.070** | 0.367 | 0.334 |
| libby | **38.89** | 29.62 | 29.35 | **0.949** | 0.819 | 0.828 | **0.095** | 0.399 | 0.342 |
| lucia | **30.90** | 26.03 | 26.63 | **0.869** | 0.690 | 0.742 | **0.178** | 0.407 | 0.329 |
| motorbike | **37.42** | 27.33 | 29.33 | **0.950** | 0.779 | 0.843 | **0.082** | 0.376 | 0.241 |
| swing | **35.70** | 26.14 | 27.88 | **0.926** | 0.722 | 0.808 | **0.119** | 0.404 | 0.289 |
| tennis | **35.65** | 27.43 | 28.81 | **0.928** | 0.806 | 0.862 | **0.120** | 0.328 | 0.209 |
| Mean | **35.35** | 27.31 | 28.09 | **0.929** | 0.763 | 0.800 | **0.098** | 0.390 | 0.302 |

## 5 Conclusion

In this work, we introduce Neural Atlas Graphs (NAGs), an editable hybrid scene representation for high-resolution learning and rendering of dynamic scenes. We find that the hybrid 2.5D representation of NAGs compares favorably in representing driving scenes. Specifically, we validate that the method achieves state-of-the-art results on the Waymo Open Dataset, with 5 dB PSNR improvement overall and 11.2 dB PSNR for rigid and 10.7 dB PSNR for non-rigid actors. We show that NAGs enable high-resolution, view-consistent environmental editing, unlocking the creation of compelling counterfactual driving scenarios and showcasing potential not only in scene arrangement and scene decomposition but also in consistent appearance editing. We also confirm the generalization of NAGs beyond driving scenes for comprehensive scene understanding and manipulation with an evaluation on the outdoor DAVIS video dataset, achieving a 7.3 dB PSNR improvement over existing methods. In the future, the differentiable representation with the low-dimensional neural atlases may allow for task-driven editing, such as learning of counterfactuals specifically to challenge a driving stack, which we exemplary demonstrated in Fig. 5.

## 6 Acknowledgments

We extend our sincere gratitude to Julian Ost, Mario Bijelic, Amogh Joshi, and William Koch from Princeton University for their insightful contributions through numerous discussions concerning the autonomous driving context, recent scene representation literature, and methodologies. This research was supported by the DFG Research Unit 5336 - Learning to Sense (Project No. 459284860) and project funding for "Polymorphic Scene Representation for Enhanced Instant Scene Reconstruction" (Project No. 510825780, Ref. KO 2960/20-1). Ilya Chugunov was supported by NSF GRFP (2039656). Felix Heide was supported by an NSF CAREER Award (2047359), a Packard Foundation Fellowship, a Sloan Research Fellowship, a Disney Research Award, a Sony Young Faculty Award, a Project X Innovation Award, an Amazon Science Research Award, and a Bosch Research Award.

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
