# OpenReview forum: "Neural Atlas Graphs for Dynamic Scene Decomposition and Editing"
_NeurIPS.cc/2025/Conference — NeurIPS 2025 spotlight_

### Official Review · Reviewer_4LSW · 2025-06-28

**Clarity:** 4
**Significance:** 3
**Originality:** 3
**Rating:** 5
**Confidence:** 3

**Summary:**

This work introduces a hybrid dynamic scene representation method called Neural Atlas Graph (NAG). It combines the advantages of two major approaches: neural scene graphs and neural atlas representations. NAG comprises nodes similar to scene graphs, but each node is represented by a neural atlas, making it more editable while maintaining visual consistency.

**Questions:**

If the authors clearly clarify the originality of the method in the relevant section and include comparisons with a SOTA atlas-based method, I would consider increasing the originality score.

If the authors answer the following questions, I would consider increasing the clarity score:

•	Is the camera pose initialized using COLMAP or another approach?

•	Are both static and dynamic objects used during initialization?

•	Are all components of a node (c_i, $\alpha_i, f_i, g_i, s_i$) optimized simultaneously according to Eq. (8), or is the rigid plane pose estimated first, followed by optimization of the other components?

•	Why are both $C_i$ (base color field) and $F_{i,c}$ (view-agnostic fields) needed? According to Eq. (5), the optimization process cannot distinguish between these two functions. Is F a learnable function while $C_i$ is not? There is no explanation of how the neural atlas map is initialized.

**Ethical Concerns:**

["NO or VERY MINOR ethics concerns only"]

**Final Justification:**

The authors provide comprehensive implementation details to ensure full reproducibility. Rather than merely merging two existing methods, they propose a novel framework that harnesses the strengths of both. These contributions are substantial and justify accepting the paper.

**Limitations:**

Yes.

**Paper Formatting Concerns:**

None.

**Quality:**

4

**Strengths And Weaknesses:**

Strengths:
- The overall performance surpasses other state-of-the-art (SOTA) methods.
- The idea is compelling, and the authors provide both mathematical details and sufficient supplementary materials.

Weaknesses:
- It is difficult to distinguish the contributions of this paper from prior work in the method section. In particular, the reviewer was confused about whether each equation is derived from previous work or newly proposed. Is the method simply a combination of existing approaches, or does it include novel contributions in the integration?
- The overall pipeline and key training details are missing, making it hard to understand the training strategy. (See questions below.)
- Although the paper combines two approaches, the experiments only include comparisons with radiance field-based methods (e.g., ORE, ERF), not with atlas-based methods.

---

> ### Author Rebuttal · Authors · 2025-07-30
>
> We appreciate the reviewer's comprehensive and constructive assessment of our paper.
>
> **1. Distinguishing Originality**
>
> The reviewer sought clarification on the originality of our method, specifically asking whether it is simply a combination of existing approaches or includes novel contributions, and requested originalty details on stated equations.
>
> We introduce NAGs as a **novel hybrid scene representation method**, positioned between Neural Scene Graphs (objects collectively form a scene graph structure) and Neural Atlases. The new representation yields an editable atlas specifically designed for video editing. While our method is inspired by various concepts in the field, our core contribution is a new representation and learning method (see Q2-Q4 below): to the best of our knowledge, there does not exist a method that models dynamic 3D scenes using 2D planar fields placed on 3D rigid motion paths in the manner we propose.
>
> **To explicitly clarify, if parts of equations are derived from others in the literature, we have explicitly cited their source directly along the equation:**
>
> - **Eq. 1:** Describes the color and alpha formation for our novel method's rendering process.
>
> - **Eq. 2:** Represents known alpha-compositing, which is standard in neural rendering (e.g., NeRFs and 3DGS), and we have explicitly cited the work of [17] for reference.
>
> - **Eq. 3 and 4:** Describe our dedicated spline-based rotation model for objects. While inspired by [6], [6] primarily focused on static binary 2D layer separation and modeled smooth camera movements, whereas our formulation is specifically adapted for per-object dynamic rigid transformations.
>
> - **Eq. 5 and 6:** Detail the color and alpha formation for every planar node. Their form, utilizing differently parameterized and regularized neural fields (one purely position-dependent, the other both position- and view-dependent) and their unique interplay, is, to the best of our knowledge, unique to our method.
>
> - **Eq. 7:** Our flow field parameterization is also inspired by [6]; however, in our method, the flow is uniquely parameterized for every object and learned within its own frame of reference, whereas [6] applies optical flow in the camera coordinate system.
>
> **2. Missing Training Details**
>
> The reviewer noted that key training details are missing. Due to NeurIPS paper size limitations, we had to move all detailed training specifics into the  **Experimental Details (Sec. 2.) of our supplement**, as also noted within our main paper. We tried to cover all information within this dedicated section, and further **have uploaded all code in an anonymous github repository** (Sec. 4.1 Ln. 245) to circumvent any missing details.
>
> **3. Comparison with Atlas-based Methods**
>
> The reviewer remarked that the experiments were missing atlas-based approaches. We would like to clarify that  *we compare against established video editing and matting methods*, addressing this point directly. As reported in **Table 3 of our main paper**, NAGs demonstrate significant performance advantages (e.g., a 7.3 dB PSNR improvement) over these baselines, specifically **Layered Neural Atlases [3] (LNA)**—a recent texture editing method—and **OmnimatteRF [4] (ORF)**—a SOTA matting approach. This validates NAGs' robust decomposition and reconstruction capabilities even without relying on explicit 3D supervision, while simultaneously enabling flexible editing. The evaluation confirms that NAGs offer substantial benefits in terms of consistency, object decomposition, and general video editing over existing methods.
>
> To further improve clarity, we will add a paragraph elaborating on these recent video matting methods and their specific usage.
>
> **Questions**
>
> **Q1**
>
> For the Waymo dataset, we rely on the camera poses provided directly by the dataset, consistent with the handling by our baselines. For the DAVIS dataset, we leverage RodynRF [7] for camera pose estimation, also consistent with our baseline OmnimatteRF [4]. Further comprehensive details regarding our camera modeling approach can also be found in the Method Details section (Section 1.1) in our supplement.
>
> **Q2**
>
> All objects are used to initialize the system, as each object has its dedicated neural fields. We leverage the input masks to initially project the color and alpha information onto our base color and alpha estimates for each object. For every object (excluding the global background), we establish its initial 3D trajectory, which will be further optimized. For the Waymo dataset, this initialization utilizes the provided 3D bounding boxes, similar to [1]. For scenes where 3D bounding boxes are not available (e.g., DAVIS), object positions are estimated based on image-to-image homographies derived from sequential frames and an initial monocular depth estimation.
>
> **Q3**
>
> Our training strategy employs a **multi-phase optimization approach** combined with a **coarse-to-fine learning strategy** to effectively optimize the various components of each node. The training is subdivided into 3 phases over 80 epochs:
>
> - 1. (Epochs 0-5): Only the positional parameters ($\mathcal{P}^{T}\_i$,$\mathcal{P}^R\_i$,$\mathcal{P}^{T}\_{cam}$,$\mathcal{P}^R\_{cam}$) are optimized to correct for initial positional errors of objects and the camera.
>
> - 2. (Epochs 5-20): The color and opacity fields ( $\mathcal{F}\_{i, c}$,$\mathcal{F}\_{i, \alpha}$) are additionally optimized, alongside the positional parameters.
>
> - 3. (Epochs 20+): All parameters ($\mathcal{P}^{T}\_i$,$\mathcal{P}^R\_i$,$\mathcal{P}^{T}\_{cam}$,$\mathcal{P}^R\_{cam}$,$\mathcal{F}\_{i, c}$,$\mathcal{F}\_{i, \alpha}$,$\mathcal{F}\_{i, f}$,$\mathcal{F}\_{i, \phi}$)  —are optimized together. Now additionally including flow, and view-dependent fields.
>
> In conjunction with these phases, we apply a coarse-to-fine learning strategy by masking encoding dimensions for the view-dependent and planar flow fields, gradually activating more encoding dimensions as training progresses. This approach limits the initial expressiveness of the view-dependent field and encourages the planar flow field to learn coarse alignments first, before enabling finer details and view dependencies to emerge, thus guiding the overall optimization process. Further Details can be found in the supplementary in Sec. 1.2 and 2.2.
>
> **Q4**
>
> Our model distinguishes between **fixed initial conditions** and **learnable neural fields**:
>
> - **Fixed Initial Conditions:** $\tilde{C}_i$ (base color) and $\tilde{A}_i$ ​ (base alpha) are **not learnable** parameters of the network. They are initialized by an initial forward projection of a reference frame onto the position-initialized plane, based on the provided masks (Sec. 3.3 Ln 181f.). This step serves to pre-condition each atlas node to a dedicated region, providing a good starting point.
>
> - **Learnable Neural Fields**: The core appearance and motion are then captured by our *learnable* neural fields: the color field $\mathcal{F}\_{i, c}$​, opacity field $\mathcal{F}\_{i, \alpha}$, flow field $\mathcal{F}\_{i, f}$, and view-dependent field $\mathcal{F}\_{i, \phi}$​. These fields are explicitly designed and optimized to serve distinct roles:
>
>   - The color field $\mathcal{F}\_{i, c}$ and opacity field $\mathcal{F}\_{i, \alpha}$ are primarily responsible for modeling the *view-agnostic, canonical appearance* of the object in its atlas space.
>
>   - The flow field $\mathcal{F}\_{i, f}$ warps this canonical appearance across time, enabling the representation of non-rigid motion and facilitating better editability by keeping the base texture consistent.
>
>   - The view-dependent field $\mathcal{F}\_{i, \phi}$ is designed to capture subtle view-dependent effects (e.g., specularities, reflections) that cannot be explained by the canonical appearance or flow alone.
>
> The optimization process distinguishes between these components through our phase-based learning and coarse-to-fine learning strategy. By initially limiting the expressiveness of the view-dependent field (via sparse encoding) and progressively activating it, we implicitly regularize the model to put the majority of the information into the non-view-dependent fields ($\mathcal{F}\_{i, c}, \mathcal{F}\_{i, \alpha}, \mathcal{F}\_{i, f}$). This forces the learnable non-view-dependent components to capture the primary appearance and motion, which are then effectively mappable by the flow for enhanced editability. The view-dependent field then only incorporates additional, subtle changes, preventing conflation and ensuring a disentangled representation.
>
> We will add these clarifications to the revised work.
>
> **References**
>
> [1] Chen, Ziyu, et al. "OmniRe: Omni Urban Scene Reconstruction." The Thirteenth International Conference on Learning Representations. 2025
>
> [2] Yang, Jiawei, et al. "EmerNeRF: Emergent spatial-temporal scene decomposition via self-supervision." arXiv preprint arXiv:2311.02077 (2023).
>
> [3] Kasten, Yoni, et al. "Layered neural atlases for consistent video editing." ACM Transactions on Graphics (TOG) 40.6 (2021): 1-12.
>
> [4] Lin, Geng, et al. "Omnimatterf: Robust omnimatte with 3d background modeling." Proceedings of the IEEE/CVF International Conference on Computer Vision. 2023.
>
> [6] Chugunov et al. “Neural Spline Fields for Burst Image Fusion and Layer Separation”. In Proceedings of the IEEE/CVF conference on computer vision and pattern recognition, pages 25763–25773, 2024.
>
> [7] Liu et al. Robust dynamic radiance fields. In Proceedings of the IEEE/CVF Conference on Computer Vision and Pattern Recognition, 2023.
>
> [17] Kopanas et al. “Neural Point Catacaustics for Novel-View Synthesis of Reflections”. ACM Trans. Graph., 41(6), 2022. Publisher: Association for Computing Machinery.

---

> > ### Comment · Reviewer_4LSW · 2025-08-05
> >
> > The authors have effectively addressed the concerns. Now, the novelty and contributions of this paper are clearly demonstrated. I will raise the originality and clarity score.

---

### Official Review · Reviewer_kRsQ · 2025-07-02

**Clarity:** 4
**Significance:** 4
**Originality:** 3
**Rating:** 5
**Confidence:** 3

**Summary:**

The paper investigates the problem of learning editable scene representations. The authors proposes a method, called Neural Atlas Graph (NAG), that combines the strengths of two prominent methods, neural scene graphs and neural atlases. The proposed method takes a video scene and the masks of foreground objects as an input, and assigns a node to each foreground object and a separate node for the background. The nodes contain information of the color, opacity and position and can model the rigid and non-rigid transformations of objects. NAG objective consists of an $\ell_1$ term for ground truth and rendered color predictions and a term that encourages the predicted opacity to match the provided coarse masks. The editing is implemented similar to the neural atlases. The authors evaluate their method for scene reconstruction and video editing. The quantitative results show significant improvements on the baselines.

**Questions:**

In general, additional experiments as explained in the quality section would be helpful. Additionally,
1. How much does the method rely on the coarse masks $\mathcal{M}$? How sensitive is the estimation of $s_i \in \mathcal{N}_i$ to the coarse masks?
2. The scenes with small ego motion were chosen for experiments. How much the output quality of the proposed and baseline methods deteriorate in scenes with higher ego motion? Could you provide sample scenes for such cases?
3. How does editing quality compare to the prior work? While the editing quality is impressive in provided videos, it would be helpful to compare it with the existing methods.

**Ethical Concerns:**

["NO or VERY MINOR ethics concerns only"]

**Final Justification:**

The authors propose an interesting method. The empirical results show significant improvements over the baselines. The paper is well-written and the authors will add additional discussion that will enhance the readability further. The authors' rebuttal was extensive and answered the concerns. The authors are also clear about the main limitations of their work.

**Limitations:**

yes

**Quality:**

3

**Strengths And Weaknesses:**

**Quality:** This paper presents a strong and methodologically sound contribution. The proposed method is clearly explained, with detailed experimental evaluations that effectively demonstrate its capabilities. The evaluation setups are transparent and reproducible, and the authors provide an anonymous codebase for reproducibility. The weaknesses could be listed as follows:
1. The number of baselines considered is relatively low compared to prior work. For instance, baseline [1] includes five different comparison methods, while this paper provides only two. Including a broader set of baselines would strengthen the empirical support. Furthermore, a brief explanation of the baseline methods—potentially in the appendix, as done in [1]—would enhance clarity and completeness.
2. The paper does not include comparisons of editing results with prior work. Baseline [1], for example, provides such results, which could offer a meaningful point of comparison.
3. The evaluation would benefit from inclusion of more challenging scenes, such as those with high ego-motion or fast-moving objects, similar to what is explored in [1] (Section D.3). This would provide additional evidence of the method’s robustness and generalizability.

**Clarity:** The paper is well-written and clearly presented. As a suggestion, adding a comparison between neural atlases, neural scene graphs, and neural atlas graphs (NAGs) could enhance the paper’s self-containment and help clarify the key ideas and design choices.

**Significance and Originality:** While the paper combines two methods in the literature, this novel fusion yields impressive results both qualitatively and quantitatively.

[1] Ziyu Chen, Jiawei Yang, Jiahui Huang, Riccardo de Lutio, Janick Martinez Esturo, Boris Ivanovic, Or Litany, Zan Gojcic, Sanja Fidler, Marco Pavone, Li Song, and Yue Wang. Omnire: Omni urban scene reconstruction. In The Thirteenth International Conference on Learning Representations, 2025.

---

> ### Author Rebuttal · Authors · 2025-07-30
>
> We sincerely thank the reviewer for the thoughtful and detailed review.
>
> **1. Number of Baselines**
>
> The reviewer considered the number of baselines low compared to prior work. We would like to clarify that we actually compared our method against four distinct baselines: OmniRe [1] (a Gaussian Splatting Method), EmerNeRF [2] (a Dynamic NeRF Method), Layered Neural Atlases [3] (a texture editing-capable method), and OmnimatteRF [4] (a recent matting method). Note that we *selected baselines for which code was available* and that ideally supported per-object removal, allowing for meaningful comparisons in our hybrid setting. We find that comparing across these diverse fields (Gaussian Splatting, Dynamic NeRF, Texture Editing, and Matting) provides strong empirical support for the broad applicability and competitive performance of NAGs, rather than limiting the comparison to a single sub-domain. Given that OmniRe demonstrated improved performance compared to the other five Gaussian Splatting methods, we focused our comparisons on it rather than including additional similar approaches. We will include a brief explanation of these baseline methods in the appendix to enhance clarity and completeness, similar to [1] and add additional automotive baselines within the revised version of our work.
>
> **2. Comparison of Editing Results to Prior Work / Question 3**
>
> The reviewer suggested including comparisons of editing results with prior work, specifically mentioning [1]. We do not find that OmniRe [1] states direct *editing* comparisons against baselines. To our knowledge, neither their main paper nor supplementary material explicitly presents quantitative or qualitative comparisons of *edited* scenes against other methods. While they do show their own scene decomposition in Figure 1 and a qualitative result comparison of the complete scene in Figure 4, these do not allow for a direct comparison of editing capabilities against baselines.
>
> In our paper, we explicitly compare the visual results of scene decomposition in Figure 3 against OmniRe on both a populated car scene and an adverse weather scene, confirming that our method excels at extracting objects and produces fewer artifacts. Furthermore, we demonstrate our scene decomposition and visual quality on two DAVIS dataset scenes in Supplementary Figure 6 (and the supplementary videos) compared against recent video editing and matting methods, again showing superior visual quality.  \
> The inherent difficulty in finding ground truth for arbitrary semantic edits (e.g., changing a car's color, adding a speed limit sign) makes direct quantitative comparisons challenging for any method in this domain.
> While we acknowledge this challenge, we have computed the Temporal (frame-by-frame) LPIPS and Fréchet inception distance (FID) [5] for our editing figures to quantify their temporal consistency and overall perceptual impact, providing an objective measure where possible. We find that the consistency is comparable to the ground truth (GT) or respective baselines.
>
> |  | T-LPIPS |  |  | FID [5] |  |  |
> |---|---:|---:|---:|---:|---:|---:|
> | Edit | Ours | ORE  [1] | GT | Ours | ORE  [1] | GT |
> | Fig. 3, Seg. 125 | 0,052 ± 0,027 | 0,054 ± 0,032 | 0,081 ± 0,033 | 2.244 ± 1.857 | 1.972 ± 1.326 | 2.228 ± 1.578 |
> | Fig. 3, Seg. 141 | 0.056 ± 0.048 | 0.063 ± 0.062 | 0.103 ± 0.072 | 0.999 ± 1.032 | 1.111 ± 1.308 | 1.517 ± 1.222 |
> | Figure 4 | 0.037 ± 0.006 | N/A | 0.043 ± 0.006 | 0.175  | N/A | 0.173 |
> | Supl. Fig. 8 | 0.249 ± 0.028 | N/A | 0.261 ± 0.026 | 0.255 | N/A | 0.185 |
> | Supl. Fig. 9 | 0.058 ± 0.012 | N/A | 0.080 ± 0.012 | 0.132 | N/A | 0.180 |
>
> For T-LPIPS we report the mean and standard deviation over all image-pairs and objects in case of Fig. 3. For FID we report the standard deviation only for the per-object decomposition evaluation, given that it is computed as a distributional measure.
>
> **3. Evaluation on Stronger Ego Motion / Question 2.**
>
> The DAVIS dataset has more camera motion than the presented Waymo sequences, but does not contain the high rotational ego motion of some autonomous driving scenes. As suggested, we add an ablation on this category of scene below:
>
> | Obj | Vehicle PSNR |  |  | Vehicle SSIM |  |  | Human PSNR |  |  | Human SSIM |  |  |
> |---: |---:|---:|---:|---:|---:|---:|---:|---:|---:|---:|---:|---:|
> | Seq. | Ours | ORE [1] | ERF [2] | Ours | ORE [1] | ERF [2] | Ours | ORE [1] | ERF [2] | Ours | ORE [1] | ERF [2] |
> | 191 | 37.60 | 30.95 | 25.81 | 0.959 | 0.936 | 0.750 | 34.40 | 25.93 | 22.91 | 0.908 | 0.742 | 0.707 |
> | 254 | 35.07 | 30.06 | 27.05 | 0.926 | 0.923 | 0.786 | 34.42 | 26.46 | 23.29 | 0.919 | 0.863 | 0.703 |
>
> | Avg | PSNR |  |  | SSIM |  |  | LPIPS |  |  |
> |---:|---:|---:|---:|---:|---:|---:|---:|---:|---:|
> | Seq. | Ours | ORE [1] | ERF [2] | Ours | ORE [1] | ERF [2] | Ours | ORE [1] | ERF [2] |
> | 191 | 32.02 | 33.22 | 29.98 | 0.892 | 0.950 | 0.872 | 0.209 | 0.090 | 0.244 |
> | 254 | 31.70 | 31.86 | 29.17 | 0.911 | 0.951 | 0.872 | 0.190 | 0.095 | 0.247 |
>
> For foreground objects, managed by dedicated nodes, our approach achieves substantial improvements (up to 8 PSNR for Vehicles/Humans) compared to all baselines. This is primarily attributed to our robust rigid motion model, which, being independent of global camera motion, is also well-suited for large ego-motion environments. The overall image scores are comparable to OmniRe and EmerNeRF (second table). As mentioned in the Limitations section, this score is not substantially higher, as we observe more artifacts in the background, as this large motion has to be generated through learning large flow vectors based on photometric loss (rather than from explicit 3D projective geometry).
>
> We emphasize that our proposed Neural Atlas Graph Model is fundamentally “2.5D” (planes + flow), with a large inductive bias which helps regularize solutions of low-parallax / sparsely observed scene elements – e.g., a vehicle that quickly drives through the scene while we only observe one side of it – but is not as well suited to model large 3D structures with large amounts of self-occlusion. However, we believe our findings in the work indicate that this is an interesting direction for future research, and hope our work can be extended as a vehicle/object-centric graph model (combining both 2D and 3D primitives in one render graph).
>
> We will add a discussion for the newly added evaluation sequences to the revised work, along with a discussion on this direction of future research.
>
> Given that we are not allowed to upload visual content in the rebuttal given the newest NeurIPS policies, we include a short text description of the added scenes:
>
> The added sequences s-1918764220984209654 (191) are showing a drivepath into a suburban pedestrian crossing, where the ego-vehicle stops shortly, vehicles and a pedestrian are crossing, after which the ego vehicle drives further into the scene.
>
> The sequence s-2547899409721197155 (254) shows the ego vehicle also on a pedestrian crossing, where a pedestrian with stroller and vehicles passing by, after which the car performs a right turn and drives into the scene, faced by a streetcar.
>
> **4. Reliance on Coarse Masks & Sensitivity**
>
> The reviewer inquired about the method's reliance on and sensitivity to coarse masks. We initialize each node's scale using the largest mask observed across all frames, incorporating an empirically found 80% margin. This margin is specifically designed to encompass object-associated effects such as shadows and reflections, which typically extend beyond the immediate object silhouette and are of low occupancy. The approach preconditions the node's spatial extent, but ensures only weak reliance on the pixel-level quality of individual frame masks for precise sizing. To further test robustness to varying input mask qualities, we will conduct experiments on perturbed masks for the final version.
>
> **5. Clarity Suggestion**
>
> The reviewer suggested adding a comparison between neural atlases, neural scene graphs, and NAGs to enhance clarity and self-containment. We will do so and add a discussion of the key ideas, design choices, strengths, and weaknesses of (Layered) Neural Atlases, Neural Scene Graphs, and Neural Atlas Graphs (NAGs).
>
> **References**
>
> [1] Chen, Ziyu, et al. "OmniRe: Omni Urban Scene Reconstruction." The Thirteenth International Conference on Learning Representations. 2025
>
> [2] Yang, Jiawei, et al. "EmerNeRF: Emergent spatial-temporal scene decomposition via self-supervision." arXiv preprint arXiv:2311.02077 (2023).
>
> [3] Kasten, Yoni, et al. "Layered neural atlases for consistent video editing." ACM Transactions on Graphics (TOG) 40.6 (2021): 1-12.
>
> [4] Lin, Geng, et al. "Omnimatterf: Robust omnimatte with 3d background modeling." Proceedings of the IEEE/CVF International Conference on Computer Vision. 2023.
>
> [5] Heusel, Martin, et al. "Gans trained by a two time-scale update rule converge to a local nash equilibrium." *Advances in neural information processing systems* 30 (2017).

---

> > ### Author Response · Authors · 2025-08-01
> > **Additional Dynamic 3DGS Baselines**
> >
> > **3. Number of Baselines (Cont.)**
> >
> > To incorporate additional baselines, we evaluated Deformable 3D Gaussians (DeformGS) [8], Periodic Vibration Gaussian (PVG) [9], and Street Gaussians (StreetGS) [10], all of which are recent dynamic 3DGS models, on our ablation sequences:
> >
> > |      | PSNR  |       |          |       |          |       | SSIM |       |          |       |          |       |
> > | -: | -: | -: | -: | -: | -: | -: | -: | -: | -: | -: | -: | -: |
> > | Seq. | Ours  | ORE [1]| DeformGS [8] | PVG [9]| StreetGS [10] | ERF [2]| Ours  | ORE [1]| DeformGS [8] | PVG [9]| StreetGS [10] | ERF [2]|
> > | 975  | 40.21 | 37.43 | 37.08    | 34.14 | 36.28    | 34.83 | 0.976 | 0.970 | 0.966    | 0.962 | 0.967    | 0.941 |
> > | 141  | 42.55 | 36.23 | 34.17    | 38.87 | 33.70    | 34.83 | 0.978 | 0.965 | 0.954    | 0.969 | 0.959    | 0.934 |
> >
> > We observe that NAGs also outperform these additional baselines in terms of PSNR and SSIM, thereby demonstrating superior performance in preserving structural details and overall image quality. We will add a full evaluation of these baselines in our revised work, including object-specific metrics on Vehicles and Humans.
> >
> > **References**
> >
> > [1] Chen, Ziyu, et al. "OmniRe: Omni Urban Scene Reconstruction." The Thirteenth International Conference on Learning Representations. 2025
> >
> > [2] Yang, Jiawei, et al. "EmerNeRF: Emergent spatial-temporal scene decomposition via self-supervision." arXiv preprint arXiv:2311.02077 (2023).
> >
> > [8] Yang, Ziyi, et al. "Deformable 3d gaussians for high-fidelity monocular dynamic scene reconstruction." Proceedings of the IEEE/CVF conference on computer vision and pattern recognition. 2024.
> >
> > [9] Chen, Yurui, et al. "Periodic vibration gaussian: Dynamic urban scene reconstruction and real-time rendering." arXiv preprint arXiv:2311.18561 (2023).
> >
> > [10] Yan, Yunzhi, et al. "Street gaussians: Modeling dynamic urban scenes with gaussian splatting." European Conference on Computer Vision. Cham: Springer Nature Switzerland, 2024.

---

> > > ### Comment · Reviewer_kRsQ · 2025-08-02
> > >
> > > I thank the authors for the detailed rebuttal and valuable contributions. The authors addressed my concerns. I raise my score correspondingly.

---

### Official Review · Reviewer_HUMq · 2025-07-02

**Clarity:** 3
**Significance:** 2
**Originality:** 2
**Rating:** 4
**Confidence:** 4

**Summary:**

This paper proposes Neural Atlas Graphs (NAGs), a novel hybrid scene representation designed for editing dynamic scenes. Unlike existing approaches that often compromise on either editability or the ability to model complex scenes, NAGs combine the strengths of both by representing scenes as a graph of moving 3D planes, with each plane corresponding to an object or the background. This representation facilitates precise 2D appearance editing as well as 3D positioning and ordering of scene elements, enabling flexible and intuitive scene manipulation. NAGs deliver state-of-the-art quantitative performance on the Waymo Open Dataset for driving scenes. Furthermore, the method demonstrates strong generalization beyond automotive contexts, achieving impressive matting and video editing results on the DAVIS dataset, which includes a diverse set of human- and animal-centric scenes.

**Questions:**

1. How does the proposed NAG approach perform when processing video inputs with significant camera motion?

2. What is the capability of NAGs for novel view synthesis with large camera displacement?

3. Can the authors explain why training NAGs for a single scene takes so long?

**Ethical Concerns:**

["NO or VERY MINOR ethics concerns only"]

**Final Justification:**

After reading the authors' rebuttal and other reviewers' opinions, most of my concerns have been addressed, and I would like to keep my positive rating.

**Limitations:**

yes

**Quality:**

3

**Strengths And Weaknesses:**

### Strengths
1. NAGs model dynamic scenes as a graph of moving 3D planes, with each plane representing an individual object or the background. This approach effectively balances the ability to handle complex, dynamic scenes with high editability, allowing for both 2D appearance modifications and 3D spatial adjustments.

3/ The qualitative results presented demonstrate robust reconstruction, decomposition, and editing of dynamic scenes, preserving scene details and enabling fine-grained edits.

### Weaknesses
1.  While NAGs provide strong editability, the practical significance of this feature in autonomous driving scenarios is uncertain. Compared to NeRF/3DGS-based approaches, NAGs’ hybrid approach may not offer a clear advantage for applications where consistent 3D rendering is critical.

2. The qualitative results lack significant camera motion, with no demonstration of novel view synthesis capabilities. This raises concerns about the method’s applicability in scenarios requiring dynamic camera perspectives. If novel view synthesis is not a priority, traditional video editing models might suffice, potentially diminishing the need for NAGs in certain use cases.

3.  The experimental evaluation omits comparisons with established image or video editing methods, particularly in tasks like matting or appearance editing where traditional methods may already perform effectively.

4. Training NAGs for a single scene requires 2 to 6 hours on an NVIDIA L40 GPU. This significant training time may pose a barrier to practical adoption.

---

> ### Author Rebuttal · Authors · 2025-07-30
>
> We thank reviewer HUMq for their analysis of and feedback on our work.
>
> **1. Practical Significance in Autonomous Driving and Comparison to NeRF/3DGS**
>
> The reviewer raises a point regarding the practical significance of Neural Atlas Graphs (NAG) editability in autonomous driving scenarios, especially compared to NeRF/3DGS-based approaches, which prioritize consistent 3D rendering. Our intent with NAGs is to offer a novel hybrid representation that excels in intuitive, object-centric manipulation, which is distinct from purely volumetric or splatting-based 3D reconstruction. While NeRF/3DGS methods excel at novel view synthesis and high-fidelity rendering from dense captures, their *texture editability is challenging and often requires re-training or complex parameter adjustments to the implicit volume or point cloud*, especially for complex non-rigid motion and semantic object manipulation.
>
> NAGs, by explicitly decomposing the scene into editable planar objects, offer a unique advantage in applications requiring semantic understanding and direct editing of texture.
>
> For a concrete application example in autonomous driving, we demonstrated in Figure 5 (supplementary material) a texture editing where we retextured the street to include a speed limit sign. As this texture editing interacts realistically with dynamic objects in the scene, forming realistic occlusions, such a technique could be highly valuable for generating realistic data for driving simulators or for training automotive image recognition agents. This demonstrates a practical application where explicit object manipulation and view-consistent texture changes are essential, complementing, rather than directly replacing existing 3D methods.
>
> We will improve the manuscript to clarify this.
>
> **2. Evaluation on Stronger Ego Motion / Question 1.**
>
> The DAVIS dataset has more camera motion than the presented Waymo sequences, but does not contain the high rotational ego motion of some autonomous driving scenes. As suggested, we add an ablation on this category of scenes below:
>
> | Obj | Vehicle PSNR |  |  | Vehicle SSIM |  |  | Human PSNR |  |  | Human SSIM |  |  |
> |-:|-:|-:|-:|-:|-:|-:|-:|-:|-:|-:|-:|-:|
> | Seq. | Ours | ORE [1] | ERF [2] | Ours | ORE [1] | ERF [2] | Ours | ORE [1] | ERF [2] | Ours | ORE [1] | ERF [2] |
> | 191 | 37.60 | 30.95 | 25.81 | 0.959 | 0.936 | 0.750 | 34.40 | 25.93 | 22.91 | 0.908 | 0.742 | 0.707 |
> | 254 | 35.07 | 30.06 | 27.05 | 0.926 | 0.923 | 0.786 | 34.42 | 26.46 | 23.29 | 0.919 | 0.863 | 0.703 |
>
> |Avg|PSNR|||SSIM|||LPIPS|||
> |-:|-:|-:|-:|-:|-:|-:|-:|-:|-:|
> |Seq.|Ours|ORE [1]|ERF [2]|Ours|ORE [1]|ERF [2]|Ours|ORE [1]|ERF [2]|
> |191|32.02|33.22|29.98|0.892|0.950|0.872|0.209|0.090|0.244|
> |254|31.70|31.86|29.17|0.911|0.951|0.872|0.190|0.095|0.247|
>
> For foreground objects, managed by dedicated nodes, our approach achieves substantial improvements (up to 8 PSNR for Vehicles/Humans) compared to all baselines. This is primarily attributed to our robust rigid motion model, which, being independent of global camera motion, is also well-suited for large ego-motion environments. The overall image scores are comparable to OmniRe (ORE) and EmerNeRF (ERF) (second table). As mentioned in the Limitations section, this score is not substantially higher, as we observe more artifacts in the background, as this large motion has to be generated through learning large flow vectors based on photometric loss (rather than from explicit 3D projective geometry).
>
> We emphasize that our proposed NAG Model is fundamentally “2.5D” (planes + flow), with a large inductive bias which helps regularize solutions of low-parallax / sparsely observed scene elements – e.g., a vehicle that quickly drives through the scene while we only observe one side of it – but is not as well suited to model complex 3D structures with large amounts of self-occlusion. However, we believe our findings in the work indicate that this is an interesting direction for future research, and hope our work can be extended as a vehicle/object-centric graph model (combining both 2D and 3D primitives in one render graph).
>
> We will add the evaluation of the newly introduced sequences into the revised work, along with a discussion on this direction of future research.
>
> As pointed out by the reviewer, our main priority is not novel view synthesis, but rather in the video editing, which we address in the following point 3.
>
> **3. Comparison to Video Editing Methods**
>
> We note, *that we did compare our method to established video matting methods* on the DAVIS dataset. Specifically, in Table 3 of our paper, our NAG demonstrates clear performance gains (7.3 dB PSNR improvement) over these video editing baselines (Layered Neural Atlases [3] (LNA) - most recent texture editing, OmnimatteRF [4] (ORF) - most recent matting), validating our ability to achieve robust decomposition and reconstruction even in the absence of 3D supervision, while simultaneously enabling flexible editing. This evaluation also confirms that even without a primary focus on novel view synthesis, NAGs offer advantages over traditional video editing approaches in terms of consistency, decomposition and editing.
>
> To make this clearer, we will add a paragraph in the revised work, elaborating on these recent video matting methods and their usage. Yet, we remain open to further baseline suggestions for our revised work.
>
> **4. Training Time / Question 3.**
>
> The reviewer finds the training time of 2 to 6 hours per scene on an NVIDIA L40 GPU as a potential barrier to practical adoption. We agree that shorter training times are highly desirable for practical applications. However, we would like to contextualize these times within the field of dynamic neural scene representation. Other state-of-the-art methods, such as OmniRe [1], also report comparable training durations (e.g., approximately one hour for a single scene at 960x640 resolution), higher on larger resolutions, while the proposed method runs for *frames with 4x more pixels.*
>
> Furthermore, highly optimized methods like Gaussian Splatting, while achieving impressive speeds, have benefited from dedicated native CUDA implementations and extensive optimization efforts over recent years [5, 6]. Below, we provide observed training times for the full resolution (1920 x 1280 for Waymo and 1920 x 1080 for DAVIS) in all experiments:
>
> | (Dataset) Method | Training Time (Minutes) |
> |-|-:|
> | (W) Ours | 140 |
> | (W) ORE [1] | 127 |
> | (W) ERF [2] | 42 |
> | (D) Ours | 198 |
> | (D) ORF [4] | 279 |
> | (D) LNA [3] | 444 |
>
> For Waymo (W), we used a subsequence from 975 with a length of 68 frames, and for DAVIS the bear sequence with 74 frames in their respective resolution. Our training time is significantly shorter than the other atlas methods on DAVIS (D). On Waymo, we are slower than the OmniRe (ORE) [1] given its efficiency in rendering. EmerNeRF (ERF) [2] is faster, given its not a scene graph method, and does not have a dedicated model per object.
>
> Below, we elaborate on three key components that contribute to our training time while being essential for a high-quality and editable NAG representation:
>
> 1. Extensive Ray-Casting: Necessary for high-resolution (up to 1920x1280), dynamic video learning, training involves extensive ray-casting (2.8×10$^8$ rays/epoch, 80 epochs). Time reduces for lower resolutions/shorter videos (<20 min).
> 2. Multi-Stage Optimization: A three-phase strategy ensures stable convergence and enhanced texture editability.
> 3. Per-Object Networks: Training time scales with scene complexity due to dedicated per-object/background neural networks.
>
> Ablation studies on different parameterizations and full training details are available in the Experimental Details (Sec. 2) of our supplementary material.
>
> **Questions**
>
> **Q1:**
> See above ("2. Evaluation on Stronger Ego Motion / Question 1.")
>
> **Q2:**
> As a *2.5D method*, NAGs are designed to infer scene structure and motion primarily from 2D observations. Consequently, novel view synthesis capabilities are inherently bound to the range of view angles seen during the fitting process. While smaller angular and positional variations of objects or the camera are possible and typically yield good results (see Fig. 4 in Supplementary Material), very large camera displacements that are significantly 'out-of-domain' of the original video cannot be handled adequately. This is because our atlas nodes are planar in 3D space, and they mimic depth parallax implicitly via flow and view-dependent fields. Therefore, large view changes can lead to perspective errors, such as looking orthogonally onto plane edges or causing objects to disappear. This limitation is inherent to our design choice: we modeled the method as 2.5D to facilitate texture edits, rather than prioritizing novel view synthesis.
>
> **Q3:**
> See above ("4. Training Time / Question 3.")
>
> **References**
>
> [1] Chen, Ziyu, et al. "OmniRe: Omni Urban Scene Reconstruction." The Thirteenth International Conference on Learning Representations. 2025
>
> [2] Yang, Jiawei, et al. "EmerNeRF: Emergent spatial-temporal scene decomposition via self-supervision." arXiv preprint arXiv:2311.02077 (2023).
>
> [3] Kasten, Yoni, et al. "Layered neural atlases for consistent video editing." ACM Transactions on Graphics (TOG) 40.6 (2021): 1-12.
>
> [4] Lin, Geng, et al. "Omnimatterf: Robust omnimatte with 3d background modeling." Proceedings of the IEEE/CVF International Conference on Computer Vision. 2023.
>
> [5] Kerbl, Bernhard, et al. "3D Gaussian splatting for real-time radiance field rendering." ACM Trans. Graph. 42.4 (2023): 139-1.
>
> [6] Wu, Guanjun, et al. "4d gaussian splatting for real-time dynamic scene rendering." Proceedings of the IEEE/CVF conference on computer vision and pattern recognition. 2024.

---

> > ### Comment · Reviewer_HUMq · 2025-08-06
> >
> > Thanks for the explanations and clarifications. After reading the authors' rebuttal and other reviewers' opinions, most of my concerns have been addressed, and I would like to keep my positive rating.

---

### Official Review · Reviewer_WUKh · 2025-07-20

**Clarity:** 4
**Significance:** 4
**Originality:** 4
**Rating:** 5
**Confidence:** 3

**Summary:**

This paper introduces Neural Atlas Graphs (NAGs), a dynamic scene representation that combines 2D neural atlases with an object-centrist 3D scene graph. Every object and also the background region is modeled as a textured planar surface with its own view-dependent neural field, optical flow, and spline-based rigid trajectory. The model is optimized at test time using only RGB photometric loss. Experiments on Waymo and DAVIS show strong performance on reconstruction metrics and demonstrate compelling view-consistent editing capabilities, including object removal, appearance changes, and motion modifications.

**Questions:**

Weaknesses
(1). The paper does not evaluate temporal consistency quantitatively (https://arxiv.org/abs/2506.01004 ), despite being a video reconstruction and editing method. A metric such as Temporal-LPIPS or inter-frame variance is necessary to support claims of stability.

(2). The evaluation is limited to Waymo and DAVIS, which are fairly structured datasets. The evaluations seem to have been done while the frame of reference is stationary - can the authors consider evaluating on datasets where the frame of reference is moving. Given it is a video based method and movement could be on the side of object as well as reference?

(3). How can the motion of the frame of reference be handled while editing a scene - such that edits are reasonably accurate and clean

(4). The editing results are purely qualitative. A stronger evaluation of performance would allow to fairly ascertain the quality of the results with confidence. Below are metrics, some of which can be used to asses the method-
    Fréchet Video Distance (FVD), Learned Perceptual Image Patch Similarity (Though it is an image-based metric, it could be be     applied frame-by-frame/or on keyframes of the edited video to assess quality and realism of the changes introduced)

   Peak Signal-to-Noise Ratio,SSIM: for evaluating specific edited regions for a comparison.

   Temporal-LPIPS - discussed in point 1

**Ethical Concerns:**

["NO or VERY MINOR ethics concerns only"]

**Final Justification:**

The responses by the authors are satisfactory. I find the work useful across multiple areas and settings in autonomous driving. I have requested the reviewers to introduce a section regarding regarding the analysis when both the object and the frame of reference are in motion and also add the code for this use-case in the final code.

**Limitations:**

see Questions.

**Quality:**

3

**Strengths And Weaknesses:**

Strengths
(1) The paper is clearly written and easy to follow, with a strong motivation.
(2) As can be seen in the supplementary video, the method effectively supports high-resolution, temporally coherent video edits using an intuitive object-wise plane representation.
(3) Strong qualitative results show convincing edits across multiple scenes and domains.
(4) The design is supported by well-structured ablation studies showing the contribution of flow, view-dependence, and trajectory modeling.

---

> ### Author Rebuttal · Authors · 2025-07-30
>
> We appreciate the thorough and constructive feedback provided by reviewer WUKh.
>
> **1. Quantitative Evaluation of Temporal Consistency**
>
> The reviewer suggested evaluating temporal consistency with metrics such as Temporal-LPIPS or Fréchet Video Distance (FVD). We report these scores in the following and will include them in the revised work, as we agree that this strengthens the overall quantitative evaluation:
>
> | (Dataset) Method | FVD ↓ | T-LPIPS |
> |---|---:|---:|
> | (W) Ours | 174 ± 238 | 0.063 ±  0.031 |
> | (W) ORE [1] | 423 ± 446 | 0.055 ± 0.029 |
> | (W) ERF [2] | 439 ± 350 | 0.053 ± 0.026 |
> | (W) GT | N/A | 0.079 ± 0.029 |
> | (D) Ours | 108 ±  35 | 0.143 ± 0.022 |
> | (D) ORF [4] | 986 ± 153 | 0.104 ± 0.023 |
> | (D) LNA [3] | 595 ±  38 | 0.116 ± 0.022 |
> | (D) GT | N/A | 0.155 ± 0.020 |
>
> In the above table, we report the FVD scores as the mean over the evaluated sequences and the Temporal (frame-by-frame) LPIPS, including their respective standard deviation. The scores have been calculated on the ablation sequences for Waymo (W) and blackswan, bear, and boat sequences for Davis (D). For FVD, lowest is best. We note that T-LPIPS is not a quality but rather a consistency metric, given its frame-by-frame application, and report the ground truth (GT) here as comparison. We find that our model outperforms baselines on FVD, for comparable image consistency scores, which aligns with the claims of the main work.
>
> We would also like to clarify that we computed the inter-frame standard deviation and reported the findings in the supplementary material (Tables 2, 3, and 4) due to space constraints in the main paper. For instance, the inter-frame standard deviation of PSNR for our method is very competitive at ±1.15 (mean PSNR 41.85), compared to OmniRe's ±2.31 (mean PSNR 36.92) and EmerNeRF's ±0.65 (mean PSNR 34.93). We will highlight these statistics in the revised main text, and further emphasize the importance of temporal consistency in addition to individual frame fidelity.
>
> **2. Evaluation on Stronger Ego Motion**
>
> The DAVIS dataset has more camera motion than the presented Waymo sequences, but does not contain the high rotational ego motion of some autonomous driving scenes. As suggested, we add an ablation on this category of scene below:
>
> | Obj | Vehicle PSNR |  |  | Vehicle SSIM |  |  | Human PSNR |  |  | Human SSIM |  |  |
> |---: |---:|---:|---:|---:|---:|---:|---:|---:|---:|---:|---:|---:|
> | Seq. | Ours | ORE [1] | ERF [2] | Ours | ORE [1] | ERF [2] | Ours | ORE [1] | ERF [2] | Ours | ORE [1] | ERF [2] |
> | 191 | 37.60 | 30.95 | 25.81 | 0.959 | 0.936 | 0.750 | 34.40 | 25.93 | 22.91 | 0.908 | 0.742 | 0.707 |
> | 254 | 35.07 | 30.06 | 27.05 | 0.926 | 0.923 | 0.786 | 34.42 | 26.46 | 23.29 | 0.919 | 0.863 | 0.703 |
>
> | Avg | PSNR |  |  | SSIM |  |  | LPIPS |  |  |
> |---:|---:|---:|---:|---:|---:|---:|---:|---:|---:|
> | Seq. | Ours | ORE [1] | ERF [2] | Ours | ORE [1] | ERF [2] | Ours | ORE [1] | ERF [2] |
> | 191 | 32.02 | 33.22 | 29.98 | 0.892 | 0.950 | 0.872 | 0.209 | 0.090 | 0.244 |
> | 254 | 31.70 | 31.86 | 29.17 | 0.911 | 0.951 | 0.872 | 0.190 | 0.095 | 0.247 |
>
> For foreground objects, managed by dedicated nodes, our approach achieves substantial improvements (up to 8 PSNR for Vehicles/Humans) compared to all baselines. This is primarily attributed to our robust rigid motion model, which, being independent of global camera motion, is also well-suited for large ego-motion environments. The overall image scores are comparable to OmniRe and EmerNeRF (second table). As mentioned in the Limitations section, this score is not substantially higher, as we observe more artifacts in the background, as this large motion has to be generated through learning large flow vectors based on photometric loss (rather than from explicit 3D projective geometry).
>
> We emphasize that our proposed Neural Atlas Graph Model is fundamentally “2.5D” (planes + flow), with a large inductive bias which helps regularize solutions of low-parallax / sparsely observed scene elements – e.g., a vehicle that quickly drives through the scene while we only observe one side of it – but is not as well suited to model complex 3D structures with large amounts of self-occlusion. However, we believe our findings in the work indicate that this is an interesting direction for future research, and hope our work can be extended as a vehicle/object-centric graph model (combining both 2D and 3D primitives in one render graph). We will add the evaluation of these sequences into the revised work, along with a discussion on this direction of future research.
>
>
> Given that we are not allowed to upload visual content in the rebuttal given the newest NeurIPS policies, we include a short text description of the added scenes:
>
> The added sequences s-1918764220984209654 (191) are showing a drivepath into a suburban pedestrian crossing, where the ego-vehicle stops shortly, vehicles and a pedestrian are crossing, after which the ego vehicle drives further into the scene.
>
> The sequence s-2547899409721197155 (254) shows the ego vehicle also on a pedestrian crossing, where a pedestrian with stroller and vehicles passing by, after which the car performs a right turn and drives into the scene, faced by a streetcar.
>
> **3. Editing under Motion**
>
> The reviewer raised a question regarding how camera motion is handled during scene editing to ensure accurate and clean edits. Edits are projected back to the planar nodes based on one or multiple edited frames, forming an **"overlay-layer" on the object plane (not the image plane),** which is therefore largely independent of the camera's ego and object-motion. However, they *are* dependent on the viewing angle with respect to the object. Our coarse-to-fine strategy aims to learn most of the view-dependent transformations in the planar flow field, which also affects the overlay-layer, resulting in deformations that can be perceived as view-consistent editing. For example, see Fig. 4., Sup. Fig. 8, 9, or the corresponding videos in the supplementary. In the boat and blackswan scene (Sup. Fig. 8, 9), the camera and object are moving independently, yet show a visually consistent texture edit.
>
> The frame of reference is used to enable intuitive editing in image space by robustly projecting edits to the planar domain via numerical inversion of the learned flow. This significantly simplifies the editing process compared to direct planar manipulation (as in [3]) and allows seamless integration with off-the-shelf image editing tools and improves editing comparability by making the texture flow-independent. We will further clarify this process in the revised work.
>
> **4. Quantitative Scores for Editing**
>
> Establishing ground truth for arbitrary edited frames on real data is challenging, making the quantitative evaluation of (texture) editing quality particularly difficult. This is a common issue in the field, and we observe that existing baseline methods [3, 4] also primarily rely on qualitative assessments. Nevertheless, we evaluated the Temporal (frame-by-frame) LPIPS and Fréchet inception distance (FID) [5] metrics to quantify the consistency and similarity of the decomposed objects, or edited frames versus their succeeding frames, for all our provided editing figures. We find that the consistency is comparable to the ground truth (GT).
>
> |  | T-LPIPS |  |  | FID [5] |  |  |
> |:---|---:|---:|---:|---:|---:|---:|
> | Edit | Ours | ORE  [1] | GT | Ours | ORE  [1] | GT |
> | Fig. 3, Seg. 125 | 0,052 ± 0,027 | 0,054 ± 0,032 | 0,081 ± 0,033 | 2.244 ± 1.857 | 1.972 ± 1.326 | 2.228 ± 1.578 |
> | Fig. 3, Seg. 141 | 0.056 ± 0.048 | 0.063 ± 0.062 | 0.103 ± 0.072 | 0.999 ± 1.032 | 1.111 ± 1.308 | 1.517 ± 1.222 |
> | Figure 4 | 0.037 ± 0.006 | N/A | 0.043 ± 0.006 | 0.175  | N/A | 0.173 |
> | Supl. Fig. 8 | 0.249 ± 0.028 | N/A | 0.261 ± 0.026 | 0.255 | N/A | 0.185 |
> | Supl. Fig. 9 | 0.058 ± 0.012 | N/A | 0.080 ± 0.012 | 0.132 | N/A | 0.180 |
>
> For T-LPIPS we report the mean and standard deviation over all image-pairs and objects in case of Fig. 3. For FID we report the standard deviation only for the per-object decomposition evaluation, given that it is computed as a distributional measure. For all stated edits, our consistency results are comparable to our reference method or the respective ground truth (GT).
>
> **References**
>
> [1] Chen, Ziyu, et al. "OmniRe: Omni Urban Scene Reconstruction." The Thirteenth International Conference on Learning Representations. 2025
>
> [2] Yang, Jiawei, et al. "EmerNeRF: Emergent spatial-temporal scene decomposition via self-supervision." arXiv preprint arXiv:2311.02077 (2023).
>
> [3] Kasten, Yoni, et al. "Layered neural atlases for consistent video editing." ACM Transactions on Graphics (TOG) 40.6 (2021): 1-12.
>
> [4] Lin, Geng, et al. "Omnimatterf: Robust omnimatte with 3d background modeling." Proceedings of the IEEE/CVF International Conference on Computer Vision. 2023.
>
> [5] Heusel, Martin, et al. "Gans trained by a two time-scale update rule converge to a local nash equilibrium." *Advances in neural information processing systems* 30 (2017).

---

> ### Comment · Reviewer_WUKh · 2025-08-04
> **Please add the "Editing under Motion " in the final Supplementary**
>
> I find the review satisfactory. I would request the authors to put special focus on "Editing under Motion" use case - where the editable objects as well as the frame of reference are in motion, analysis in the supplementary. The real world use cases would strongly demand this feature. Adversarial robustness is a field where motion is important while dealing with adversaries. A detailed analysis of this direction is important for a strong real-world applicability of the research.
>
> It is also expected that the code for this scenario be made available in the final code.

---

> > ### Author Response · Authors · 2025-08-04
> > **"Editing under Motion " in the final Supplementary**
> >
> > We agree and will add an expanded analysis of the mentioned editing scenario to the supplementary of our revised work. In addition, we will provide a dedicated code tutorial in our final code base to enhance usability.

---

### Note · Authors · 2025-08-12

Following the discussion, reviewers recognized the paper’s clarity, compelling qualitative results, well-structured ablation studies, and the novelty of **Neural Atlas Graphs (NAGs)** as an object-centric representation for dynamic scenes. The main points raised were addressed in the rebuttal with the following additions:

- **Complex Motion**: New experiments on scenes with stronger ego-motion demonstrate that the motion model remains effective, particularly for foreground objects.
- **Quantitative Metrics**: Additional temporal consistency (FVD, Temporal-LPIPS) and editing quality (FID, Temporal-LPIPS) metrics confirm competitive performance.
- **Expanded Baselines**: Comparisons with recent dynamic 3DGS baselines (DeformGS, PVG, StreetGS) show improved results.

These updates, along with further stated clarifications and discussion during the rebuttal, will be included in the revised version and are intended to fully address the feedback provided during the review and discussion.

---

### Decision · Program_Chairs · 2025-09-17

**Decision:**

Accept (spotlight)

**Comment:**

The paper received three Accept and one Borderline Accept. All the reviewers appreciate that this paper proposes a novel object-centric representation for dynamic scene decomposition and editing, with a reasonable technical approach, good evaluation, and some interesting conclusions for the community. Thus, this paper achieves the NeurIPS acceptance requirement. However, AC highly urges the authors to go through the detailed comments carefully to polish the writing and provide extra information for comparison and completeness, so as to make sure the final acceptance.